# Preferences for public engagement in decision-making regarding four COVID-19 non-pharmaceutical interventions in the Netherlands: A survey study

Sophie Kemper[1,2]*, Marloes Bongers[1], Frank Kupper[2], Marion De Vries[1], Aura Timen[1,2,3]

1 National Coordination Centre for Communicable Disease Control, National Institute for Public Health and the Environment, Bilthoven, The Netherlands, 2 Athena Institute, Faculty of Science, VU University Amsterdam, Amsterdam, The Netherlands, 3 Primary and Community Care, Radboud University Medical Center, Nijmegen, The Netherlands

* Sophie.kemper@rivm.nl

## Abstract

### Background

Worldwide, non-pharmaceutical interventions (NPIs) were implemented during the COVID-19 crisis, which heavily impacted the daily lives of citizens. This study considers public perspectives on whether and how public engagement (PE) can contribute to future decision-making about NPIs.

### Methods

An online survey was conducted among a representative sample of the public in the Netherlands from 27 October to 9 November 2021. Perceptions and preferences about PE in decision-making on NPIs to control COVID-19 were collected. Preferences regarding four NPIs were studied: Nightly curfew (NC); Digital Covid Certificate (DCC); Closure of elementary schools and daycares (CED); and physical distancing (1.5M). Engagement was surveyed based on the five participation modes of the IAP2 Spectrum of Public Participation, namely inform, consult, advice, collaborate and empower.

### Results

Of the 4981 respondents, 25% expressed a desire to engage in decision-making, as they thought engagement could improve their understanding and the quality of NPIs, as well as increase their trust in the government. Especially for the NPIs DCC and NC, respondents found it valuable to engage and provide their perspective on trade-offs in values (e.g. opening up society versus division in society by vaccination status). Respondents agreed that the main responsibility in decision-making should stay with experts and policy-makers. 50% of respondents did not want to engage, as they felt no need to engage or considered themselves insufficiently knowledgeable. Inform was deemed the most preferred mode of engagement, and empower the least preferred mode of engagement.

**Data Availability Statement:** All relevant data are within the paper and its Supporting information files. Minor changes have been made to the

underlying dataset to secure the privacy of our respondents.

**Funding:** The authors received no specific funding for this work.

**Competing interests:** The authors have declared that no competing interests exist.

## Conclusion

We reveal large variations in public preferences regarding engagement in NPI decision-making. With 25% of respondents expressing an explicit desire to engage, and considering the benefit of PE in other areas of (public) health, opportunities for PE in NPI decision-making might have been overlooked during the COVID-19 pandemic. Our results provide guidance into when and how to execute PE in future outbreaks.

## Introduction

Worldwide, non-pharmaceutical interventions (NPIs) were implemented in order to minimize transmission of the SARS-CoV-2 virus and control the COVID-19 pandemic [1]. Examples of NPIs are physical distancing, nightly curfews and travel restrictions [2]. Even though NPIs are necessary to protect the health of citizens, they can drastically affect people's daily lives. It is likely that millions of citizens have been affected by NPIs in terms of their livelihoods, mental well-being and social life [3]. The implementation of NPIs led to public discussion and discontent, which occasionally evolved into demonstrations and protests [4, 5]. For example, in the Netherlands, riots started after the government imposed a three-week partial lockdown in November 2021 [6]. During the same period, in Rome, citizens gathered to protest after COVID entry passes were issued for work, venues and public transport [7]. Similar events occurred in other countries. These widespread protests might indicate that policies regarding NPIs were not always in line with the needs and preferences of the public.

In most countries, governments have the ultimate decision-making power to implement NPIs, and are advised by national expert panels [8–10]. Whether or not NPIs reduce COVID-19 transmission depends on the degree of citizen compliance. Support for NPIs could be increased by integrating the public perspective more directly into the decision-making process, an approach that has become more prevalent in recent decades. The public is comprised of separate individuals, who express different opinions and interest when confronted with an issue. The public is not a monolithic entity, but is comprised of people with a diverse range of demographic, epidemiologic, social and economic characteristics. Rowe and Frewer (2005) defined public engagement as: "The practice of involving members of the public in the agenda-setting, decision-making, and policy-forming activities of organizations or institutions responsible for policy development". In general, three rationales for public engagement (PE) in decision-making have been suggested in the literature [11, 12]. First, engaging the public in high-impact decisions can be viewed as a democratic right, which could increase the legitimacy of decisions. Second, by engaging in the decision-making process, citizens can gain insight into how decisions are made. This could improve understanding, transparency and trust. Subsequently, this could enhance acceptance of policy decisions, and lead to higher compliance [10]. Third, the public can bring in new perspectives on science and policy discourse by raising concerns or suggesting new ideas that may otherwise be overlooked. This could increase the feasibility and quality of decisions [13, 14]. PE in decision-making can be executed in various modes, which are context-specific. These five modes are inform, consult, advice, collaborate and empower. The first mode is inform, in which the public is provided with accurate information. The fifth mode is empower, in which final decision-making is placed in the hands of the public [15]. Another factor in PE is the type of citizen to engage; for example PE could focus on citizens representative of the general population, or on population groups with certain characteristics, such as vulnerable population groups who were (disproportionally) impacted by NPIs or COVID-19.

There are several examples of successful PE in NPI decision-making. For example in Central Vietnam, a new Dengue control method was studied, during which island residents, experts, local leaders and governmental staff were consulted regarding their questions, concerns and acceptance of the method. This resulted in a culturally appropriate consultation and communication process and more support for the control method [16]. In another study, young men and women were consulted about communication efforts regarding Zika virus in the United States Virgin Islands. With their input, a culturally relevant communication campaign was implemented [17]. In addition, during the COVID-19 pandemic, efforts were made to gather perspectives and preferences among citizens regarding NPIs. For example, studies from the UK, Italy, Germany and China collected the preferences of random samples of the general population on strategies to manage COVID-19 [18–21]. Another survey reported the preferences of citizens from thirteen countries regarding COVID-19 vaccine allocation [22]. In the Netherlands, throughout the COVID-19 pandemic, a behavioral unit carried out a multitude of studies on public opinions regarding NPIs. These findings were translated into recommendations to improve policy and communication [23].

The studies outlined above aimed to collect data on public perspectives and create recommendations for decision-making accordingly. This data can be used as an inspiration for more participatory decision-making in the field of epidemic management. However, there are fewer examples of actual implementation of the public perspectives more directly into NPI decision-making, in order to create a more participatory decision-making process. Despite the promising potential of PE on the legitimacy, acceptability and quality of decision-making, little is known about how PE could be best implemented to be valuable for epidemic management. A first step in gaining a better understanding of how to implement PE successfully in epidemic management, and specifically NPI decision-making, is to determine whether and how citizens would want to engage. This raises several important questions regarding citizens' views on how and when citizens should engage, which topics citizens should engage in, and who should be responsible for final decisions. Answering these questions could be an important step for developing future engagement practices in NPI decision-making that more closely align with the needs and interests of the public. We will therefore address the following research question:

*What are the preferences of Dutch citizens about public engagement in decision-making regarding non-pharmaceutical interventions to control the COVID-19 epidemic in the Netherlands?*

## Materials and methods

### Study population and procedure

An online survey was conducted between 27 October and 9 November 2021. Around this time in the Netherlands, infections and hospitalizations increased due the fourth COVID-19 wave, driven by the Delta variant. Subsequently, more stringent NPIs were implemented at the start of November 2021 [24]. The survey was sent to members of an online research panel (I&O Research Panel; ISO/IEC 20252). This panel consisted of more than 36.000 members of 18 years and older from the Netherlands, who were invited for membership based on random samples of name and address data. A subset of 11.505 members was invited to participate, which was representative of the general Dutch population in terms of gender, age, place of residency, education level, and migration background. The survey was piloted with 10 cognitive interviews to ensure clarity and understanding. Prior to participation, respondents were

informed about the study's goal, duration, usage of data, privacy, and the possibility of withdrawal. Informed consent was given at the start of the survey. The duration of the survey was approximately 15 minutes. Respondents who completed the survey earned points, which could later be exchanged for a gift card or a donation to charity (standard procedure panel organization).

The study protocol was approved by The Centre for Clinical Expertise at the National Institute for Public Health and the Environment (study protocol number LCI-498).

## Conceptual framework and survey design

A comprehensive survey on engagement in COVID-19 decision-making was created based on multiple frameworks and findings from an earlier study on the same topic (specific information about this can be found in S2 File) [25]. Four NPIs implemented in the Netherlands were used as a starting point, and were based on the framework by Hoppe and Hisschemöller 1993 depicted in Fig 1 [26]. In this framework, four problem types are described. Hoppe and Hisschemöller define a problem as a construct, that comprises both values and facts. These values and facts stand at the root of how a problem is categorized. As depicted in Fig 1, on the y-axis the level of certainty about relevant knowledge about the problem is represented. On the x-axis, the level of certainty about relevant norms and values is represented. In this study, four NPIs are identified and categorized based on the framework. By using four NPIs that fit the four problem types, information regarding a wide variety of NPI-types could be raised:

A = Nightly curfew (NC). This NPI was implemented from 23 January to 28 April 2021 in the Netherlands, as the outbreak situation was highly concerning. Much backlash from citizens arose as it highly interfered with values like freedom and autonomy. Moreover, scientific knowledge regarding the effectiveness of NCs was scarce [27]. Due to these characteristics, NC was categorized as type A.

B = Closure of elementary schools and daycares (CED). This NPI was implemented between 16 March and 6 April 2020 in the Netherlands, as part of the intelligent lockdown. CED was categorized as type B, as it was unknown how CED would impact the epidemic. However,

**Fig 1. Four problem types in policy design and analysis by Hoppe and Hisschemöller 1993, with corresponding NPIs and date of implementation.**

many citizens expressed concerns about the safety of teachers and daycare employers and children when keeping schools and daycares open. Hereafter, the government decided to employ the CED [27].

C = Digital Covid Certificate for events (DCC). At the start of July 2021, during a very short period, this NPI was implemented. It was more or less clear that such a certificate would decrease transmission, however, there was much public debate regarding this NPI. In particular, debate about the division in society on vaccination status [28]. Therefore, DCC was categorized as type C.

D = 1.5meter social distancing (1.5M). Since March 2020 a number of basic measures were implemented. One of these basic measures was the 1.5 meter social distancing. Social distancing was an evidence-based measure to reduce transmission, and was generally well accepted by citizens [29]. Due to these characteristics, 1.5M was categorized as type D.

A clear context was outlined per NPI with dates and epidemiological information regarding COVID-19. Following the description of each NPI, identical questions were asked for every NPI, which can be divided into five themes related to PE in the decision-making process (see Table 1 for more information). The five themes were based on multiple frameworks, such as the IAP2 Spectrum of Public Participation, the Risk Analysis Framework and the Emergency Management Cycle [15, 30, 31]. Furthermore, general literature on PE and findings of an earlier study on the same topics were used [25]. Additional information regarding the setup of the survey can be found in S2 File.

Theme 1—desire for engagement & reasons (not) to engage; respondents were asked *if* they would have wanted to engage in NPI decision-making and *why*.

Theme 2—mode of engagement; respondents were asked *how* they would have wanted to engage, by using the IAP2 Spectrum of Public Participation [15]. This spectrum comprises five modes of PE ranging from *Inform*; which provides the public with balanced and objective information, to *Empower*; which places the final decision making power in the hands of the public. With each mode, the level of impact of the public on decisions increases.

Theme 3—phase in decision-making process; respondents were asked *which phase of the NPI decision- making process* they would have wanted to engage in.

Theme 4—timing of engagement; respondents were asked *when* (which phase of the outbreak) they would have wanted to engage.

Theme 5—responsibilities; respondents were asked *who* they thought should have been engaged, who has the responsibility for decision-making, and to what extent contributions of the public should be mandatorily incorporated into decision-making.

Engagement was explained as "having a say or participating in decision-making about how to manage the COVID-19 epidemic. Your opinions, ideas and experiences will be used to create and design the management of COVID-19." The complete survey can be found in S3 File. Every respondent received questions regarding only two out of the four NPIs to avoid overburdening the respondent. These two NPIs were randomly allocated in order to ensure there were respondents with similar demographic characteristics in every subgroup. In total, six possible combinations of NPI-sets were answered: (1) NC-CED, (2) NC-DCC, (3) NC-1.5M, (4) CED-DCC, (5) CED-1.5M, and (6) DCC-1.5M.

## Statistical analysis

Data was analyzed using IBM SPSS Statistics V28. Respondents who completed the survey in four minutes or less were excluded from the analysis. In addition, data was checked for cases

**Table 1. Content of the survey, displaying main themes, variables, corresponding survey questions, type of question and answer categories.**

| THEME | VARIABLE | SURVEY QUESTION | TYPE OF QUESTION AND ANSWER CATEGORIES |
|---|---|---|---|
| 1. DESIRE FOR ENGAGEMENT | | In general, would you have like to be engaged in the decisions regarding [NPI*]? | Multiple Choice Question, single answer. Yes; No; I don't know. |
| **1. REASONS TO ENGAGE | Understanding | I did want to be engaged because it would help me better understand how [NPI] was created. | Likert-scale. 1 = not true, 5 = true. |
| | Anxiety | I did want to be engaged because it would decrease the overall anxiety that I have regarding the COVID-19 epidemic. | |
| | Trust | I did want to be engaged because it would increase my trust in the government. | |
| | Acceptability | I did want to be engaged because it would increase my adherence to [NPI]. | |
| | Quality | I did want to be engaged because it would increase the quality of [NPI] overall. | |
| **1. REASONS NOT TO ENGAGE | Lack of knowledge | I did not want to be engaged because I have too little knowledge about [NPI]. | Likert-scale. 1 = not true, 5 = true. |
| | Lack of time | I did not want to be engaged because I have too little time. | |
| | Lack of need | I did not want to be engaged because I don't feel the need to. | |
| | No direct effect | I did not want to be engaged because [NPI] didn't affect me directly. | |
| 2. MODE OF ENGAGEMENT | Inform | You will receive all information about the considerations regarding the NPI, and how the final decision is made. | Ranking question. Rank the 5 modes of engagement in order from least suitable (1) to most suitable (5) according to how you would have liked to be engaged in the NPI. |
| | Consult | Your opinion will be asked about certain questions or problems regarding the NPI. Your opinion will be taken under consideration. | |
| | Advice | Your advice is asked regarding all steps in the decision-making process, and will be certainly used in the final decision. | |
| | Collaborate | You (together with a group of citizens) will collaborate with the government about all the decisions regarding NPI. | |
| | Empower | You (together with a group of citizens) have the final decision-making power regarding NPI. The government will support you. | |
| 3. PHASE IN DECISION-MAKING PROCESS | Situation assessment | Would you have liked to be engaged in assessing the severity of the outbreak situation before introducing [NPI]? | Likert-scale. 1 = certainly not, 5 = most certainly. |
| | Effect of [NPI] | Would you have liked to be engaged in determining the effect of the [NPI]? | |
| | Trade-off between interests | Would you have liked to be engaged in thinking about the trade-offs between the interests regarding [NPI]? | |
| | Practicability | Would you have liked to be engaged in thinking about how to best implement [NPI]? | |
| | Communication | Would you have liked to be engaged in thinking about the communication regarding [NPI]? | |
| 4. TIMING OF ENGAGEMENT | Timing of engagement during COVID-19 epidemic | When do you think was the best time to be engaged in [NPI]? | Multiple Answer Question. Before the outbreak; During the outbreak; After the outbreak; Never; Other*** |

(*Continued*)

**Table 1.** (Continued)

| THEME | VARIABLE | SURVEY QUESTION | TYPE OF QUESTION AND ANSWER CATEGORIES |
|---|---|---|---|
| 5. RESPONSIBILITIES | Mandatory incorporation | To what extent do you think that your and other citizens' contributions should be included in the final decision? | Likert-scale. 1 = voluntarily incorporation, 5 = mandatory incorporation. |
| | Who | Who do you think should have been engaged in [NPI]? | Multiple Answer Question. All citizens; Signing up; Only organizations; Only representative persons; Representative sample; None; Other.**** |
| | Responsibility of politicians | How much responsibility should politicians have had regarding [NPI] decisions? | Likert-scale. 1 = no responsibility at all, 5 = complete responsibility. |
| | Responsibility of experts | How much responsibility should experts have had regarding [NPI] decisions? | |
| | Responsibility of citizens | How much responsibility should citizens have had regarding [NPI] decisions? | |

\* Please note, every question displayed was repeated for all four NPI (nightly curfew, closure of elementary schools and daycares, Digital Covid Certificate for events and 1.5 meter social distancing). Therefore, in this table [NPI] is used as generic term.

\*\*Routing question: If respondents answered yes to desire for engagement, they had to provide answers about reasons to engage. If respondents answered no to desire for engagement, they had to provide answers about reasons not to engage.

\*\*\* Full answer options: (1) Before the outbreak. We did not know whether the NPI would be implemented, but we would have clarity about the perspective of the public regarding the NPI; (2) During the outbreak, when it became clear that the NPI was necessary; (3) After the outbreak, during evaluation. This information can be used in future epidemics; (4) Never; (5) Other.

\*\*\*\* Full answer options: (1) All citizens had to be engaged; (2) Only citizens who wanted to engage should have been able to sign up for engagement; (3) Only certain organizations and/or corporations that were involved with the NPI, such as associations for healthcare organizations; (4) Only persons representing interest groups in society, such as a leader of a youth organization; (5) A representative panel of the Dutch population that was chosen based on certain demographic characteristics such as age and gender; (6) None of the above; (7) Other.

of straight lining, which was defined as giving identical answers to a minimum of three questions, three times in a row [32]. Descriptive analysis (frequencies and proportions) were carried out for all variables, each based on single survey questions. Each respondent answered questions about two NPIs and, due to the randomization of the NPIs, six possible combinations of NPI-sets emerged. Differences in responses to the four NPIs were analyzed. Per set, paired t-tests were executed for Likert-scale data, Mcnemar tests were executed for Multiple Answer Question data (multiple answers) and $\chi^2$ tests were executed for Multiple Choice Question data (single answer) with more than two categories. Multiple testing correction with the Benjamini-Hochberg procedure and a false discovery rate of 5% was applied for all aforementioned tests [33]. Finally, multinominal regression for each NPI was executed to study differences in desire for engagement (dependent) based on gender, age, education level, place of residency and migration background (independent). The aim of the multinominal regression was to possibly identify differences in desire for engagement between population groups. These differences might be important to take into account when shaping future engagement practices.

## Results

In total, 5008 persons completed the survey (response rate of 43.5%). No cases of straight lining were found, and 17 respondents (0.38%) were excluded due to a response time ≤4 minutes. This resulted in a sample population of 4981 persons, whose characteristics are displayed in Table 2. The sample population accurately reflected most demographic distributions in the Dutch population. However, persons in the age categories of 18–24 and 35–49

**Table 2. Distribution of gender, age category, education level, region of residency and migration background of all respondents and per subgroup.** The distribution of these demographic variables of the total population in the Netherlands in 2021 is displayed as a reference.

| | TOTAL | | SUBGROUPS* | | | | | | *REFERENCE*: DISTRIBUTION OF POPULATION NETHERLANDS** |
|---|---|---|---|---|---|---|---|---|---|
| | | | NC-DCC | NC-CED | NC-1.5M | DCC-CED | DCC-1.5M | CED-1.5M | |
| | n | % | % | % | % | % | % | % | % |
| GENDER | | | | | | | | | |
| FEMALE | 2534 | 50.9 | 50.8 | 51.0 | 52.6 | 48.2 | 53.0 | 49.5 | 50.6 |
| MALE | 2447 | 49.1 | 49.2 | 49.0 | 47.4 | 51.8 | 47.0 | 50.5 | 49.4 |
| AGE CATEGORY | | | | | | | | | |
| 18–24 | 278 | 5.6 | 6.4 | 6.0 | 5.8 | 5.6 | 4.7 | 5.0 | 10.9 |
| 25–34 | 756 | 15.2 | 13.1 | 14.9 | 15.9 | 16.8 | 14.9 | 15.6 | 16.0 |
| 35–49 | 886 | 17.8 | 18.9 | 18.8 | 18.9 | 17.5 | 15.3 | 17.3 | 23.3 |
| 50–64 | 1813 | 36.4 | 36.7 | 35.1 | 35.2 | 36.9 | 38.8 | 35.8 | 26.1 |
| 65+ | 1248 | 25.1 | 25.0 | 25.3 | 24.1 | 23.2 | 26.3 | 26.3 | 23.6 |
| EDUCATION LEVEL | | | | | | | | | |
| LOW | 1113 | 22.3 | 21.1 | 21.6 | 22.9 | 21.8 | 23.7 | 23.0 | 20.6 |
| MIDDLE | 1891 | 38.0 | 40.6 | 38.6 | 37.7 | 37.9 | 37.2 | 35.7 | 39.6 |
| HIGH | 1977 | 39.7 | 38.3 | 39.9 | 39.4 | 40.3 | 39.0 | 41.3 | 39.8 |
| REGION OF RESIDENCY | | | | | | | | | |
| WEST | 2217 | 44.5 | 42.6 | 45.0 | 46.1 | 45.4 | 41.2 | 46.7 | 45.5 |
| NORTH | 512 | 10.3 | 11.2 | 9.8 | 10.8 | 10.6 | 10.3 | 8.9 | 10.0 |
| EAST | 997 | 20.0 | 21.9 | 20.4 | 18.1 | 19.0 | 21.5 | 19.2 | 20.8 |
| SOUTH | 1246 | 25.0 | 24.2 | 24.7 | 24.6 | 24.9 | 26.6 | 25.2 | 23.7 |
| *MISSING* | 9 | | *1* | *1* | *3* | *1* | *3* | | |
| MIGRATION BACKGROUND | | | | | | | | | |
| NATIVE DUTCH | 4503 | 90.4 | 90.3 | 88.0 | 90.7 | 92.2 | 90.8 | 90.5 | 76.9 |
| NON-WESTERN MIGRANT | 156 | 3.1 | 2.5 | 4.3 | 3.1 | 3.2 | 2.8 | 2.9 | 12.3 |
| WESTERN MIGRANT | 322 | 6.5 | 7.1 | 7.7 | 6.2 | 4.6 | 6.4 | 6.7 | 10.8 |
| TOTAL NUMBER OF RESPONDENTS | 4981 | | 864 | 835 | 841 | 784 | 817 | 840 | |

* Each respondent answered questions about two NPIs, which were randomly allocated. this led to six subgroups within the sample population. The subgroups were used to analyse differences between NPis. NC = nightly curfew, Dcc = digital covid certificate, ced = closure of elementary schools and daycares, 1.5m = 1.5meter social distancing.

** Reference is based on population numbers of 2021 from the national statistics office in the netherlands.

were underrepresented. In addition, persons with a migrant background were also underrepresented. The subgroups had comparable demographic distributions to the sample population.

## 1. Desire for engagement

For NC, 27% of respondents *did want to engage* in the decision-making, 49% *did not want to engage* and 24% of respondents *did not know or were neutral* (see Table 3). For CED, 19% *did want to engage*, 59% *did not* and 22% *did not know or were neutral*. For DCC, 30% *did want to engage* in the decision-making, 46% *did not*, and 24% *did not know or were neutral*. Lastly, for

**Table 3. Summary table of results for theme 1,2,4 and 5.** Data is presented as proportions (%), mean scores on Likert-scales (with standard deviations).

| QUESTION | (ANSWER) CATEGORIES | NIGHTLY CURFEW | CLOSURE OF ELEMENTARY SCHOOLS AND DAYCARES | DIGITAL COVID CERTIFICATE FOR EVENTS | 1.5M SOCIAL DISTANCING |
|---|---|---|---|---|---|
| | | PERCENTAGE OF ALL RESPONDENTS | | | |
| 1. DESIRE FOR ENGAGEMENT | YES | 27% | 19% | 30% | 21% |
| | NO | 49% | 59% | 46% | 52% |
| | I DON'T KNOW / NEUTRAL | 24% | 22% | 24% | 26% |
| | | MOST CHOSEN MODE OF ENGAGEMENT IN RANK* (% OF ALL RESPONDENTS) | | | |
| 2. SUITABLE MODE OF ENGAGEMENT (1 = LEAST, 5 = MOST) | RANK 1 | EMPOWER (72%) | EMPOWER (71%) | EMPOWER (70%) | EMPOWER (73%) |
| | RANK 2 | COLLABORATE (44%) | COLLABORATE (43%) | COLLABORATE (45%) | COLLABORATE (46%) |
| | RANK 3 | ADVICE (47%) | ADVICE (49%) | ADVICE (47%) | ADVICE (50%) |
| | RANK 4 | CONSULT (41%) | CONSULT (44%) | CONSULT (42%) | CONSULT (44%) |
| | RANK 5 | INFORM (46%) | INFORM (49%) | INFORM (46%) | INFORM (50%) |
| | | PERCENTAGE OF ALL RESPONDENTS | | | |
| 4. TIMING OF ENGAGEMENT | BEFORE THE OUTBREAK | 21% | 21% | 19% | 19% |
| | DURING THE OUTBREAK | 51% | 48% | 60% | 60% |
| | AFTER THE OUTBREAK | 31% | 30% | 27% | 27% |
| | NEVER | 14% | 19% | 15% | 14% |
| | OTHER | 2% | 2% | 2% | 2% |
| | | MEAN (SD) | | | |
| 5. RESPONSIBILITIES (1 = NO RESPONSIBILITY AT ALL) | POLITICIANS | 3.9 (SD = 1.0) | 3.9 (SD = 1.0) | 3.9 (SD = 1.0) | 3.9 (SD = 1.0) |
| | EXPERTS | 3.9 (SD = 0.8) | 4.0 (SD = 0.8) | 3.9 (SD = 0.9) | 4.0 (SD = 0.8) |
| | CITIZENS | 2.6 (SD = 1.1) | 2.6 (SD = 1.0) | 2.7 (SD = 1.1) | 2.7 (SD = 1.1) |
| 5. INCORPORATION (1 = VOLUNTARY) | INCORPORATION OF CONTRIBUTIONS OF CITIZENS | 2.7 (SD = 1.3) | 2.6 (SD = 1.2) | 2.8 (SD = 1.3) | 2.7 (SD = 1.3) |
| | | PERCENTAGE OF ALL RESPONDENTS | | | |
| 5. WHO | ALL CITIZENS | 12% | 7% | 14% | 18% |
| | SIGNING UP | 24% | 20% | 27% | 25% |
| | ORGANIZATIONS AND COMPANIES | 51% | 63% | 58% | 44% |
| | PERSONS REPRESENTING INTEREST GROUPS | 27% | 42% | 32% | 27% |
| | REPRESENTATIVE SAMPLE OF DUTCH POPULATION | 32% | 20% | 33% | 31% |
| | NONE | 12% | 8% | 9% | 13% |
| | OTHER | 7% | 7% | 6% | 7% |

* THIS RANKING QUESTIONS WAS ANSWERED FROM LEAST SUITABLE (RANK 1) TO MOST SUITABLE (5).

1.5M, 21% *did want to engage*, 52% *did not* and 26% *did not know or were neutral*. The differences in percentages of respondents who *did want to engage* in decision-making between NC (27%), CED (19%), DCC (30%) and 1.5M (21%) were significant (q<0.01; q represents the corrected p-value by using the Benjamini-Hochberg method). Similarly, there were significant differences between all four NPIs (q<0.01) regarding the percentages of respondents who *did not want to engage*, *did not know or were neutral*. Detailed results of the subgroups analysis can be found in S4 File.

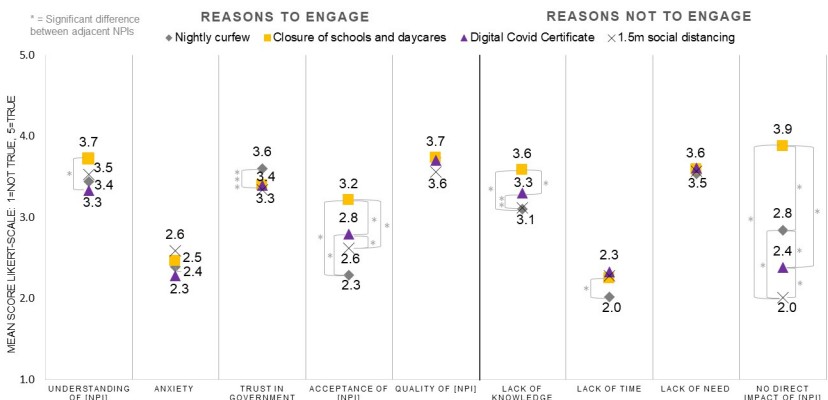

**Fig 2. Five possible reasons for respondents to engage displayed on the left side of the figure, and four possible reasons for respondents not to engage displayed on the right side of the figure.**

Fig 2 displays the findings regarding the reasons why respondents desired to engage. The questions on reasons to engage were only answered by respondents who did want to engage (routing question). When the mean value of this variable scored a 3 or higher, it was interpreted as a true reason for respondents (not) to engage. For NC, respondents perceived that engaging in decision-making would have slightly *increased their overall understanding of NC* (M = 3.4), their *trust in the government* (M = 3.6), and *the quality of NC* (M = 3.7). Respondents did not think that engaging would *decrease their overall anxiety about the outbreak* (M = 2.4) or *increase their acceptance of NC* (M = 2.3). For CED, DCC, and 1.5M, respondents also perceived that engaging in NPI decision-making would have slightly *increased their overall understanding of the NPI*, *trust in government*, and *the quality of the NPI*. Similarly, respondents did not think that engaging would *decrease their overall anxiety about the outbreak*, or *increase their acceptance* of CED, DCC, or 1.5M.

On the right side of Fig 2, the reasons why respondents desire not to engage are displayed. The questions on reasons not to engage were only answered by respondents who did not want to engage (routing question). For NC and 1.5M, respondents indicated that *a lack of need for engagement* was an important reason not to engage in decision-making (M = 3.5 and M = 3.6). For NC and 1.5M, respondents considered *a lack of time* and *the NPI not having a direct impact on their lives* no true reasons not to engage. For CED and DCC, respondents indicated that *a lack of knowledge* and *a lack of need for engagement in general* were reasons not to engage. *A lack of time* was again not considered a reason not to engage. For CED, respondents indicated that *the NPI not having a direct impact on them* was a reason for them not to engage (M = 3.9), which had a significantly higher mean compared to the means of the other NPI (q<0.01).

## 2. Mode of engagement

The survey questions of themes 2–5 were answered by all respondents. For NC, the most suitable mode of engaging the public in decision-making (rank 5) was *Inform*, according to 46% of the respondents (see Table 3). According to 41% of the respondents, the second most suitable mode of engagement (rank 4) was *Consult*. *Advice* was mostly placed on rank 3 by 47% of the respondents. *Collaborate* was mostly deemed as second least suitable mode of engagement (rank 2) by 44% of respondents. The least suitable mode of engagement (rank 1) was *Empower*, according to 72% of the respondents. This ranking was more or less similar for CED, DCC

and 1.5M. As for these NPIs, *Inform* was also chosen as the most suitable mode of engagement, followed by *Consult* (rank 4), *Advice* (rank 3) and *Collaborate* (rank 2). For all three NPIs, *Empower* was deemed least suitable mode of engagement by the majority of the respondents (see Table 3). Detailed results of the analysis can be found in S5 File.

### 3. Phase in decision-making process

As displayed in Fig 3, for all four NPIs, the respondents had a neutral-to-negative disposition towards engagement in all five phases in the decision-making process. When the mean value of this variable scored a 3 or higher, it was interpreted as the respondents having a positive disposition towards engagement. For NC, *Trade-offs between interests*, had the highest mean on the Likert-scale (2.8) out of all five phases. This phase comprises the weighing of interests, principles and values when implementing NPIs. For CED and DCC, *Trade-offs between interests* also had the highest means (2.7 and 2.9) compared to the four other phases in decision-making. For 1.5M, *Practicability* had the highest mean (2.6). *Practicability* refers to the practical implementation and execution of NPIs.

### 4. Timing of engagement

Respondents were asked about their preferences on the best moment to engage in decision-making. For these questions, it was possible to give multiple answers. For NC, the best time to engage according to the respondents was *during the outbreak*, followed by *after the outbreak*, *before the outbreak*, *never* and *other* (see Table 3). This order of preferences of timing of engagement was similar for CED, DCC and 1.5M: first *during the outbreak*, followed by *after the outbreak*, *before the outbreak*, *never* and *other*.

### 5. Responsibilities

Respondents were asked who they thought should have been engaged in decision-making. It was possible to give multiple answers to this question (see Table 3). For NC, respondents mostly preferred engaging *Organizations with interest in NC*, followed by *A representative sample of the Dutch population*, *Persons representing interests groups in society*, *Only citizens who want to sign up*, *All citizens*, *None*, and *Other*. For DCC and 1.5M, identical sequences of answers was found. For CED, the most preferred group to engage was also *Organizations with*

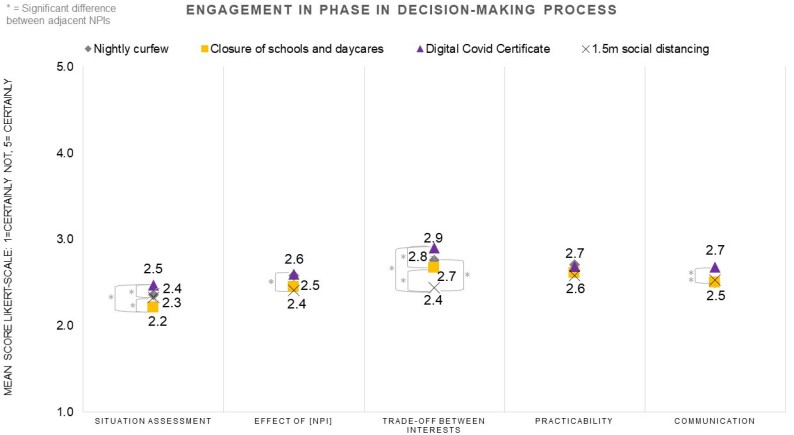

**Fig 3. Attitudes of respondents towards engagement in the five phases of the decision-making process per NPI.**

*interest in CED*. However, the sequence of the other categories differed compared to the other NPIs: *Persons representing interests groups in society* was seen as second most preferred category, followed by *A representative sample of the Dutch population*, Only citizens who want to sign up, *None*, *All citizens*, and *Other*.

For all four NPIs, respondents believed that both experts and policymakers should have *a lot of responsibility* regarding decision-making, with mean values of 3.9 and 4.0 for experts, and 3.9 for policy-makers (see Table 3, 1 = no responsibility). For all NPIs, respondents believed that the public should have between *a little* and *medium responsibility*, with means of 2.6 and 2.7. As such, the opinion of the respondents regarding this variable barely differed across all NPIs. Furthermore, for all four NPIs, respondents believed that the *incorporation of contributions of citizens* should be kept voluntarily, with mean values between 2.6 and 2.8 (using a score of 3 as a cut-off point between voluntary and mandatory incorporation).

### Desire for engagement and demographic variables

Our data suggests that men wanted to engage in decision-making significantly more than women for NC (OR = 1.63 p<0.01), DCC (OR = 1.50 p<0.01), and 1.5M (OR = 1.49 p<0.01). For CED, no significant predictive value for gender for desire to engage was found. Furthermore, for all four NPIs, age also appears to be a predictor for desire to engage; the higher the age group of the respondents, the more likely they would want to engage in decision-making. For DCC for example, 65+ year old respondents were 3.23 (p<0.01) times more likely to prefer engagement over no engagement compared to 18–24 years old. Similarly, when comparing 65 + years old to 25–34 years old, the older group was 2.38 (p<0.01) times more likely to prefer engagement over no engagement. No significant predictive values were found regarding education level, place of residency, migration background and desire to engage (see S6 File).

### Discussion

The aim of this study was to gain insights into the preferences of Dutch citizens on public engagement (PE) in decision-making regarding non-pharmaceutical interventions (NPIs), as these insights could contribute to the legitimacy, quality, and public acceptance of NPIs that were implemented to control the COVID-19 epidemic. Our study reveals various preferences for engagement in NPI decision-making among the Dutch population. For each of the four NPIs included in our study, approximately half of the respondents had no desire to engage in decision-making, as they did not feel a need to engage, or considered themselves insufficiently knowledgeable about the subject. In line with this, the majority of respondents indicated that they thought experts and government policy makers should be responsible for decision-making. Furthermore, about a quarter of the respondents did not know or were neutral towards engagement. A quarter of the respondents did want to engage. Our results therefore suggest that despite varying preferences regarding engagement, there are a considerable number of citizens who do want to engage. Those who wanted to engage expected that engagement could increase their understanding of NPIs, raise their trust in the government, and improve the quality of the NPIs.

The desire for engagement among our respondents was the highest for the NPIs which restrict freedom of movement, namely NC and DCC. In the framework of Hoppe and Hisschemöller, these NPIs were categorized as "problems" without consensus on relevant norms and values (Fig 1). The results suggest that insights from the public could be beneficial, especially for NPIs for which there is no public consensus. For such NPIs, PE could have the benefit of informing the public and improving understanding of the implications of NPIs. Furthermore, a more active mode of engagement like consulting the public, could have benefit in

improving the quality of NPIs. PE could be particularly valuable in epidemics, in which high-stakes and complex decisions have to be made under time pressure, within a context of resource scarcity and eroding public trust in decision-makers [34–36]. Desire for engagement was lowest for 1.5M and CED. For CED, this can be explained by the lack of direct impact on daily life, as not all citizens have (pre) school-aged children who attend elementary school or daycare. The impact of a decision on people's daily lives is already a known prerequisite for desire for engagement [37]. For 1.5M, an explanation for the finding that there was a lower desire for engagement could be that social distancing had already been established as an evidence-based prevention measure [38].

The results for other preferences for engagement (themes) revealed few differences between the four NPIs. For example, only slight differences were found between the NPIs regarding the percentage of respondents who ranked Inform as the most suitable mode of engagement (between 46–50%). Inform was judged the most preferred mode of engagement, and was defined as receiving all information about the considerations regarding the final decision about implementing the NPI. This finding could signify a need among the public for more transparency and insights into the considerations made during COVID-19 management. Such transparency could positively influence preventive behavior during a pandemic and trust in authorities [39]. The other half of respondents preferred the other modes of engagement as suitable, for example, Consult was mostly placed as the second most suitable mode of engagement, followed by Advice as the third most suitable. This reveals a desire among citizens to have more impact on decisions, rather than only being informed appropriately. Moreover, our respondents indicated that, during PE, policy-makers should carefully consider contributions of the public, but should not be obliged to adopt them. If not adopted, however, it is crucial for them to explain why. Similarly, another study identified that incorporating public perceptions in healthcare decision-making should be kept voluntarily. Additionally, they identified that when contributions are not adopted, it is vital to explain why [40]. Empower was considered the least suitable mode of engagement. This corresponds with our other results that indicate citizens prefer experts and policy-makers to have more responsibility than citizens, and that contributions of citizens should not be mandatorily incorporated into decision-making.

Half of our respondents expressed no desire to engage. It is possible that some of these respondents are not yet aware of the potential their contributions may have in regard to decision-making, which is a barrier that has frequently been described in engagement literature [41, 42]. A majority of the respondents underlined this with a preference to engage targeted groups such as organizations or representative samples of the population. The engagement of such subgroups can also be an appropriate form of engagement, as the perspectives of the public can be assessed on a community level, instead of an individual level [40, 43]. In addition, most respondents indicated that they do not want to engage (or had neutral dispositions towards engagement) in any of the phases of the decision-making process. This could be explained by the majority of respondents only wanting to be informed instead of having a more active role in e.g. assessing the severity of the outbreak situation. These results raise the question of how many citizens are required to engage in order to justify implementing PE in practice (and conversely, how many citizens should not want to engage to justify no implementation). Moreover, besides "group size", other considerations could also be important, if not more important, regarding PE. Other considerations could include be how much impact decisions have on citizens, available resources such as time, and the need for diverse perspectives, as well as the exact payoff or impact of PE in public health.

According to our respondents, engagement should take place as soon as it becomes clear that NPIs are necessary. This means engagement should be incorporated directly into the

decision-making process, rather than during the preparedness or evaluation phases. This is in line with other recommendations on PE, which state that PE should be executed upstream, within the planning process of a decision, in order to truly establish co-creation between the public and policy-makers [44]. However, PE should not be done too early in the process, as one might feel no compelling reason to engage at that early stage [45]. Overall, for future research, we recommend using these findings (in addition to previous research on the implementation of PE) to shape and apply engagement in NPI decision-making, and evaluate these practices.

## Limitations

PE and/or NPI decision-making might have been a complex or abstract topic for respondents. We carefully attempted to create a comprehensive survey by testing and improving it with cognitive interviews. Furthermore, we added supporting information and news articles in the survey to clarify the context. During data collection, of the four NPIs, only the 1.5M was implemented in real time. This means that for NC, CED and DCC, respondents had to reflect on experiences in the past, whereas for the 1.5M people could actively reflect on PE in decision-making. In addition, the epidemic situation was rapidly evolving during our study. For instance, more stringent NPIs were implemented when we performed our data collection. This dynamic situation could have affected the views of the respondents regarding engagement in NPI decision-making. In line with this, the findings are context specific for the COVID-19 epidemic in the Netherlands. Moreover, the sample population, though carefully sampled to reflect the Dutch population, underrepresented people between the age of 18 to 24 years old and 35 to 49 years old, as well as people with a western and non-western migration background. It is therefore uncertain if our findings are also applicable to these populations. Another factor is the distinction between intention to engage, which we identified in this study, and actual real-time engagement when the opportunity arises. It is uncertain to what extent people are actually willing to participate in real-time engagement efforts [46]. As a final point, the public is comprised of people with a diverse range in characteristics. In this study, we have taken a first step in gaining insight into the preferences of the public. However, it would be valuable to gain more in-depth insight into differences in preferences in various groups within the public. This variety could be explored through demographic characteristics, as well as positions on or viewpoints about the COVID-19 pandemic; for example, this could be done by collecting data on the views of vulnerable populations or people with a lack of trust in the government.

## Conclusions

Our results suggest a large variation in preferences regarding engagement in NPI decision-making to control COVID-19 among the Dutch population. Given the potential benefit of PE, there are opportunities that might have been overlooked during the decision-making process for specific NPIs that have been implemented in the past. These include, for example, the nightly curfew and the Digital Covid Certificate. Our results suggest that informing the public, being more transparent regarding the decision-making process, and maybe having more active modes of engagement, could have benefited COVID-19 management. In addition, our study provides guidance regarding when and how it may be preferable for the public to engage during epidemics. Understanding these preferences may help decision-makers to develop better engagement practices for specific groups in the population, which may ultimately enhance their ability to improve the control of COVID-19 and possible future crises.

## Supporting information

**S1 File. Information about the four non-pharmaceutical interventions (NPIs).**
(DOCX)

**S2 File. Rationales behind the setup of the survey.**
(DOCX)

**S3 File. Complete survey.**
(DOCX)

**S4 File. Results of subgroup analysis.**
(DOCX)

**S5 File. Detailed results on suitable mode of engagement (Theme 2).**
(DOCX)

**S6 File. Results of multinominal linear regression analysis.**
(DOCX)

## Acknowledgments

First, we would like to thank all the participants that made this study possible. Furthermore, many thanks to Doret de Rooij, Sandra Kamga Kengne and Jacobiene Janse for their valuable feedback during this study.

## Author Contributions

**Conceptualization:** Sophie Kemper, Marloes Bongers, Frank Kupper, Aura Timen.

**Formal analysis:** Sophie Kemper, Marion De Vries.

**Funding acquisition:** Aura Timen.

**Investigation:** Sophie Kemper.

**Methodology:** Sophie Kemper, Marloes Bongers, Frank Kupper, Aura Timen.

**Resources:** Aura Timen.

**Supervision:** Marloes Bongers, Frank Kupper, Marion De Vries, Aura Timen.

**Validation:** Sophie Kemper, Aura Timen.

**Writing – original draft:** Sophie Kemper.

**Writing – review & editing:** Marloes Bongers, Frank Kupper, Marion De Vries, Aura Timen.

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
