## [Decision Letter · Decision Letter 0]

28 Mar 2023

PONE-D-22-27762Preferences for public engagement in decision-making regarding four COVID-19 non-pharmaceutical interventions in the Netherlands: a survey studyPLOS ONE

Dear Dr. Kemper,

Thank you for submitting your manuscript to PLOS ONE. After careful consideration, we feel that it has merit but does not fully meet PLOS ONE’s publication criteria as it currently stands. Therefore, we invite you to submit a revised version of the manuscript that addresses the points raised during the review process.

We look forward to receiving your revised manuscript.

Kind regards,

Ali B. Mahmoud, Ph.D.

Academic Editor

PLOS ONE

4. Please amend your manuscript to include your abstract after the title page.

Reviewers' comments:

Reviewer's Responses to Questions

**Comments to the Author**

1. Is the manuscript technically sound, and do the data support the conclusions?

Reviewer #1: Yes

Reviewer #2: Yes

2. Has the statistical analysis been performed appropriately and rigorously? 

Reviewer #1: Yes

Reviewer #2: Yes

3. Have the authors made all data underlying the findings in their manuscript fully available?

Reviewer #1: Yes

Reviewer #2: Yes

4. Is the manuscript presented in an intelligible fashion and written in standard English?

Reviewer #1: Yes

Reviewer #2: No

5. Review Comments to the Author

Reviewer #1: • A very important topic

• Reasonably excellent sample size of respondents

• More elaborations are needed on the (engagement in decision-making) types. Usually many government did not declare the (specifics) of who is most affected by COVID (i.e., age composition, more elderly’s, etc.).

• I the (Introduction) part, more details (and references) should be provided from (other countries) on the issue of (engaging the public). The practice of more countries should be included. Moreover, some more elaborations are needed on the (type of engagement) and how (candid) the interactions are.

• In the (study population) part, a relatively (short) time interval (period 27 October and 9 November 2021- driven by the Delta 114 variant) was mentioned. Any specific elaborations on why this time. Would the timing effect the results?

• In the (statistical analysis) part, it would be nice if you could provide the (Cronbach reliably) of each dimension, when possible. This could provide more (reliability) to the instrument used.

• In the (Results) section, more explanations of Table 2 is recommended.

• The (discussion) part needs to be elaborated upon more. It should cover (all) results obtained from the study. More comparison is also needed with other similar research with respect to results.

• In the (limitation) part, would you have recommended different analysis methods in future studies?

Reviewer #2: The manuscript would greatly benefit from a careful grammar check, particularly in the introduction and discussion sections.

Abstract

Line 14-15: Suggest revising to “whether and how public engagement (PE) may play an important role in the development of future NPIs.”

Closure of elementary schools and daycares is abbreviated as “CSD” in the abstract, but is “CED” in the remainder of the manuscript. Please fix.

Line 22: Add abbreviation for 1.5m social distance for consistency.

Line 28-29: Suggest removing “The public could play a role ... during decision-making” and edit to “Despite the desire for engagement, respondents agreed that the responsibility should stay with experts and policy-makers.”

Discuss how Inform was the major mode of engagement that respondents wanted within the results section of the abstract. While Empower was the least suitable mode.

Please expand on the Conclusion aspect of the abstract.

Introduction

Line 98: Expand to make sure it’s clear you mean “methods to use to engage the public”

Lines 100-103 needs wordsmithing to improve readability

Materials and methods

Expand and explain as to how each of the selected NPIs fit within the problem types.

Lines 133-135: This needs wordsmithing for clarification

Results

In general, when discussing the different topics within each answer, I suggest keeping consistent with how they are formatted within the results section. In some instances the topics are italicized when discussed (Line 231: Inform), sometimes they are italicized and within single quotation marks (Line 241: ‘Trade-offs between interests’), and sometimes they are just within single quotation marks (Line 242: ‘Practicability’). Standardizing how these are presented will enhance readability and make it clear to the reader that a topic within a theme is being discussed.

EX: Line 248-249: “during the outbreak, when it became clear that the NPI was necessary” may read as two choices when it is in fact one choice.

EX: Line 241: ‘Trade-offs between interests’ vs. Line 242: ‘Practicability’

Lines 201-207: Please make sure the percentages as they are listed in the text align with what is demonstrated in Table 3 – Many percentages do not match up. Please fix as necessary.

EX:

- Line 202: CED, 19% did want to engage

- Line 203 DCC, 30% did want to engage

- Line 205-206: who did want to engage in decision making between ... CED (30%), DCC (19%)...

- According to Table 3: CED – 19%; DCC – 30%

For evaluations of the Likert scales, I suggest adding a brief sentence before explaining the results to describe how you interpreted the mean values of the respondents (like did you set the midway point as 2.5 and thus any mean score above that would be agreeable vs. below that would be not agreeable?). This simple introductory sentence will help your future readers who may not be as familiar with how such data is interpreted.

Lines 230-235: I suggest you should also comment on the results (as shown in Table 3) that demonstrate the least suitable mode of engagement being Empower. This shows that in general, the public just wants more information concerning the NPI decision-making process, but does not think that the decision-making process should be put in the hands of the general public.

Also who was answering the “least suitable mode of engagement” question? The people who said they did want to engage or did not want to engage? This needs to be made clear.

Line 258: Should “Organizations with interest in NC” be changed to “Organizations with interest in CED” instead?

You do not discuss the results of 5. INCORPORATION. I would discuss it (even if briefly) or remove it from Table 3.

For the “Phase in Decision-making process” – you report a neutral-to-negative response. Could this be due the majority of respondents indicating that their preferred mode of engagement is “Inform” while the survey questions in the “Phase in decision-making process” are phrased in ways that the public appears to have a more active role rather than just being informed? (i.e. “engaged in assessing the severity of the outbreak” “engaged in determining the effect”)

Lines 277 and 279: Can you frame the likelihood statistics for preferring engagement among age groups in the inverse? (i.e. 65+ year old respondents were 3.22 times more likely to prefer engagement over no engagement in comparison to 18-24 year olds). It says the same thing, but the fold increase appears more meaningful in this manner.

Have you done any comparative statistics concerning the demographic variables and the survey themes/topics? If so, were there any findings of interest? I wonder if some of these demographic variables also have an impact on the survey responses.

Discussion

Line 314: Replace “extra” with “additional”

Line 312-315: Expand on what you mean by valuable (EX in the sense of government trust, understanding of NPIs, etc.). Also would it be new insights? Your survey results say that most respondents just wanted additional information, not necessarily to provide input. If you re-phrase this in the sense of the results suggesting that provision of more information concerning the NPI design and execution could be valuable for the public, I think that would be more in agreement with your results.

Line 315: “made under high-pressure time constraints”

Line 317-319: Your data provides evidence for this, as the mean value for CED “no direct impact of NPI” within the reasons to not engage section of Figure 2 is 3.9. I would refer to this here.

Line 331-332: Where is this shown? Is this the interpretation of the results from 5. INCORPORATION? If so, you for sure want to make sure that this is discussed in the results section, otherwise these results are coming out of nowhere, without any previous discussion or context.

Line 333-335: At the same time, you have results that say over 70% of respondents say empower is the least suitable mode of engagement... I think these results are more telling of public opinion, and is more in line with the general support for experts and policy-makers in the decision-making process. It might be worthwhile to comment on both findings (i.e. empower being the most important mode after Inform, but empower also being voted the least appropriate mode).

Line 346: “A quarter of... desire to engage” I would remove this sentence – it does not really add anything to the discussion where it currently is.

Line 347-351: I think another thing to consider is how PE is expected to impact/benefit NPIs. For example, while you have increased understanding of NPIs and trust of government, but among those that did want engagement, you had relatively neutral mean scores from respondents regarding increased acceptance/adherence of NPI. So if you put all this effort into public engagement, but without the public health payoff of increased NPI adherence, would PE actual be beneficial?

- If there are any statistics concerning NPI adherence within the Netherlands, the addition of this information would give important context for the paper and its results. This can be added to the introduction section.

Limitations

Line 370: What is it meant by “put theory into practice when it comes to ... citizenship”?

6. PLOS authors have the option to publish the peer review history of their article (what does this mean?). If published, this will include your full peer review and any attached files.

Reviewer #1: **Yes: **Masood Badri

Reviewer #2: No

---

## [Author Response · Author response to Decision Letter 0]

23 May 2023

May 2023

Dear Dr. Ali B. Mahmoud and reviewers,

We are very happy to know that you are interested in our manuscript and the possibility to have it published in PLOS ONE. Many thanks to you and the two reviewers for providing us with valuable feedback to improve the quality of our manuscript. After carefully considering all comments, we have created an improved manuscript accordingly. Below, you will find our point-by-point response to all comments, and corresponding changes in the manuscript and where to find them (the page and line numbers used below refer to the clean version of the manuscript). 

Furthermore, we have added the minimal data set underlying to our results. We have altered two variables in order to ensure the anonymity of our respondents (in collaboration with data stewards). First, in the dataset, we only provide data in three age categories instead of five categories. Second, we only provide data regarding whether a respondent has a migration background yes or no, instead of subdividing migration background into (1) Native Dutch, (2) Non-western migrant and (3) western migrant. We hope this is sufficient.

We sincerely hope that our revised manuscript can be accepted for publication in PLOS ONE. All authors have seen and approved the revised manuscript.

Sincerely, on behalf of all the co-authors.

Reviewer 1

Comment 1: More elaborations are needed on the (engagement in decision-making) types. Usually many government did not declare the (specifics) of who is most affected by COVID (i.e., age composition, more elderly’s, etc.).

Response: Revised as suggested. We added more information in the introduction paragraph about the various types of engagement, and which type of population groups could be engaged in NPI decision-making in general (page 3 line 73-79):

‘’PE in decision-making can be executed in various modes, which are context-specific. These five modes are inform, consult, advice, collaborate and empower. The first mode is inform, in which the public is provided with accurate information. The fifth mode is empower, in which final decision-making is placed in the hands of the public (15). Another factor in PE is the type of citizen to engage; for example PE could focus on citizens representative of the general population, or on population groups with certain characteristics, such as vulnerable population groups who were (disproportionally) impacted by NPIs or COVID-19.’’

Comment 2: I the (Introduction) part, more details (and references) should be provided from (other countries) on the issue of (engaging the public). The practice of more countries should be included. Moreover, some more elaborations are needed on the (type of engagement) and how (candid) the interactions are.

Response: Revised as suggested. We have elaborated in more detail upon the setting, engagement type and type of citizen in the studies referred to in the introduction. For all studies, we have reflected upon what ultimately happened with the input of the engaged citizens. For some studies it resulted in culturally appropriate campaigns that were actually implemented, but for other studies, as to our knowledge, the input of citizens was just used for research purposes. We included examples from Vietnam, United States Virgin Island, UK, Italy, Germany, China, and one study including findings from 13 countries (Autralia, Brazil, Canada, Chile, China, Colombia, France, India, Italy, Spain, Uganda, UK and US). (page 3-4 line 81-93)”:

‘’There are several examples of successful PE in NPI decision-making. For example, McNaughton et al. (2014) consulted island residents, experts, local leaders and governmental staff regarding their questions, concerns and acceptance of a new Dengue control method in central Vietnam. This resulted in a culturally appropriate consultation and communication process and more support for the control method (16). In another study, Brittain et al. (2019) consulted young men and women about communication efforts regarding Zika virus in the United States Virgin Islands. With their input, a culturally relevant communication campaign was implemented(17). In addition, during the COVID-19 pandemic, efforts were made to gather perspectives and preferences among citizens regarding NPIs. For example, studies from the UK, Italy, Germany and China collected the preferences of random samples of the general population on strategies to manage COVID-19(18, 19, 20, 21) Another survey by Duch et al. (2021) reported the preferences of citizens from thirteen countries regarding COVID-19 vaccine allocation (22). In the Netherlands, throughout the COVID-19 pandemic, a behavioral unit carried out a multitude of studies on public opinions regarding NPIs. These findings were translated into recommendations to improve policy and communication (23).’’

Comment 3: In the (study population) part, a relatively (short) time interval (period 27 October and 9 November 2021- driven by the Delta 114 variant) was mentioned. Any specific elaborations on why this time. Would the timing effect the results?

Response: Thank you, this is a good point. The COVID-19 pandemic was a highly dynamic period. We planned our data collection during the pandemic period to gain insight into preferences of the public while their memories regarding the NPIs were still fresh. The timing of our data collection was not influenced by the specific epidemic situation at that time (increasing hospitalizations due to fourth COVID-19 wave, driven by Delta-variant). We acknowledge that the epidemic situation could have influenced the results on which we reflect in our limitation chapter (page 18 line 393-399):

‘’During data collection, of the four NPIs, only the 1.5M was implemented in real time. This means that for NC, CED and DCC, respondents had to reflect on experiences in the past, whereas for the 1.5M people could actively reflect on PE in decision-making. In addition, the epidemic situation was rapidly evolving during our study. For instance, more stringent NPIs were implemented when we performed our data collection. This dynamic situation could have affected the views of the respondents regarding engagement in NPI decision-making. In line with this, the findings are context specific for the COVID-19 epidemic in the Netherlands.’’

Comment 4: In the (statistical analysis) part, it would be nice if you could provide the (Cronbach reliably) of each dimension, when possible. This could provide more (reliability) to the instrument used.

Response: In our analysis, we did not create constructs from multiple variables, therefore we deem reliability analysis not a suitable method to apply. All statistical analyses were done on variables consisting of the responses to single survey questions. We clarified this by amending Table 1 (showing all variables and corresponding survey questions) and by rewording the text under ‘Statistical analysis’ in the methods section (page 9 line 200-202):

‘’Descriptive analysis (frequencies and proportions) were carried out for all variables, each based on single survey questions.’’

Comment 5: In the (Results) section, more explanations of Table 2 is recommended.

Response: Revised as suggested. We have added an explanation about the emergence and the use of subgroups in Table 2 (page 10): 

‘’Each respondent answered questions about two NPIs, which were randomly allocated. This led to six subgroups within the sample population. The subgroups were used to analyse differences between NPIs. NC = Nightly curfew, DCC = Digital Covid Certificate, CED = Closure of elementary schools and daycares, 1.5M = 1.5meter social distancing.’’

Furthermore, we have altered the title of the Table to provide more clarity (page 10):

‘’Table 2. Distribution of gender, age category, education level, region of residency and migration background of all respondents and per subgroup. The distribution of these demographic variables of the total population in the Netherlands in 2021 is displayed as a reference.

Furthermore, we have added a sentence about the demographic distribution of the subgroups in the text (page 9 line 220):

‘’The subgroups had comparable distributions in demographics to the sample population.’’

Comment 6: The (discussion) part needs to be elaborated upon more. It should cover (all) results obtained from the study. More comparison is also needed with other similar research with respect to results.

Response: Revised as suggested. We have included the results of all topics obtained from the study. First, we added the results regarding the phases in the decision-making process into the discussion (page 17 line 370-376):

‘’In addition, most respondents indicated that they do not want to engage (or had neutral dispositions towards engagement) in any of the phases of the decision-making process. This could be explained by the majority of respondents only wanting to be informed instead of having a more active role in e.g. assessing the severity of the outbreak situation. These results raise the question of how many citizens are required to engage in order to justify implementing PE in practice (and conversely, how many citizens should not want to engage to justify no implementation)’’

Furthermore we have included the results of the topic about voluntary/mandatory incorporation of contributions of citizens in the final decision in the discussion, and subsequently expanded our discussion section with comparable literature (page 16 line 355-362):

‘’Moreover, our respondents indicated that, during PE, policy-makers should carefully consider contributions of the public, but should not be obliged to adopt them. If not adopted, however, it is crucial for them toto explain why. Similarly, Litva et al (2002) identified that incorporating public perceptions in healthcare decision-making should be kept voluntarily. Additionally, they identified that when contributions are not adopted, it is vital to explain why (40). Empower was considered the least suitable mode of engagement. This corresponds with our other results that indicate citizens prefer experts and policy-makers to have more responsibility than citizens, and that contributions of citizens should not be mandatorily incorporated into decision-making.’’

And the following text (page 17 line 365-370):

‘’Half of our respondents expressed no desire to engage. It is possible that some of these respondents are not yet aware of the potential their contributions may have in regard to decision-making, which is a barrier that has frequently been described in engagement literature (41, 42). A majority of the respondents underlined this with a preference to engage targeted groups such as organizations or representative samples of the population. The engagement of such subgroups can also be an appropriate form of engagement, as the perspectives of the public can be assessed on a community level, instead of an individual level (40, 43).‘’

Comment 7: In the (limitation) part, would you have recommended different analysis methods in future studies?

Response: This is a good point, we have added a part at the end of the discussion about future research. We think that future research should focus on putting the findings of this study (and preceding studies) into practice. This means experimenting with various types of engagement practices with various types of population groups and evaluating the process and results of these efforts (page 17 line 386-388):

‘’Overall, for future research, we recommend using these findings (in addition to previous research on the implementation of PE) to shape and apply engagement in NPI decision-making, and evaluate these practices.’’ 

Reviewer 2

Comment 1: The manuscript would greatly benefit from a careful grammar check, particularly in the introduction and discussion sections.

Response: Revised as suggested. We asked the language editing service of our University (VU University Amsterdam) for a grammar. These changes can be checked in the document titled ‘’Revised Manuscript with Track Changes’’. 

Abstract

Comment 2: Line 14-15: Suggest revising to “whether and how public engagement (PE) may play an important role in the development of future NPIs.”

Response: Partly revised as suggested (page 1 line 15-16).

Comment 3: Closure of elementary schools and daycares is abbreviated as “CSD” in the abstract, but is “CED” in the remainder of the manuscript. Please fix.

Response: Revised as suggested (page 1 line 22).

Comment 4: Line 22: Add abbreviation for 1.5m social distance for consistency.

Response: Revised as suggested (page 1 line 22).

Comment 5: Line 28-29: Suggest removing “The public could play a role ... during decision-making” and edit to “Despite the desire for engagement, respondents agreed that the responsibility should stay with experts and policy-makers.”

Response: Revised as suggested (page 2 line 30-31).

Comment 6: Discuss how Inform was the major mode of engagement that respondents wanted within the results section of the abstract. While Empower was the least suitable mode.

Response: Revised as suggested. Next to adding this to the results section, we have also added using the five participation modes to the methods section (page 1 line 23-24):

Methods

‘’Engagement was surveyed based on the five participation modes of the IAP2 Spectrum of Public Participation, namely inform, consult, advice, collaborate and empower.’’

And (page 2 line 33-34):

Results

‘’Inform was deemed the most preferred mode of engagement, and empower the least preferred mode of engagement.’’ 

Comment 7: Please expand on the Conclusion aspect of the abstract.

Response: Revised as suggested. We have altered the conclusion part in order for it to better reflect the main conclusions in our manuscript (page 2 line 37-40):

‘’We reveal large variations in public preferences regarding engagement in NPI decision-making. With 25% of respondents expressing an explicit desire to engage, and considering the benefit of PE in other areas of (public) health, opportunities for PE in NPI decision-making might have been overlooked during the COVID-19 pandemic. Our results provide guidance into when and how to execute PE in future outbreaks.’’

Introduction

Comment 8: Line 98: Expand to make sure it’s clear you mean “methods to use to engage the public”

Response: Revised, we have altered this paragraph in the introduction as a whole (page 4 line 100-109):

‘’A first step in gaining a better understanding of how to implement PE successfully in epidemic management, and specifically NPI decision-making, is to determine whether and how citizens would want to engage. This raises several important questions regarding citizens’ views on how and when citizens should engage, which topics citizens should engage in, and who should be responsible for final decisions. Answering these questions could be an important step for developing future engagement practices in NPI decision-making that more closely align with the needs and interests of the public. We will therefore address the following research question:

What are the preferences of Dutch citizens about public engagement in decision-making regarding non-pharmaceutical interventions to control the COVID-19 epidemic in the Netherlands?’’

Comment 9: Lines 100-103 needs wordsmithing to improve readability

Response: Revised as suggested. We have made alterations in this paragraph to improve clarity. We focused on the expectation of citizens regarding whether they would want to engage, how and when and the responsibilities of experts, citizens and policy-makers during PE processes (pg 4 line 99-109): 

‘’Despite the promising potential of PE on the legitimacy, acceptability and quality of decision-making, little is known about how PE could be best implemented to be valuable for epidemic management. A first step in gaining a better understanding of how to implement PE successfully in epidemic management, and specifically NPI decision-making, is to determine whether and how citizens would want to engage. This raises several important questions regarding citizens’ views on how and when citizens should engage, which topics citizens should engage in, and who should be responsible for final decisions. Answering these questions could be an important step for developing future engagement practices in NPI decision-making that more closely align with the needs and interests of the public. We will therefore address the following research question:

What are the preferences of Dutch citizens about public engagement in decision-making regarding non-pharmaceutical interventions to control the COVID-19 epidemic in the Netherlands?’’

Materials and methods

Comment 10: Expand and explain as to how each of the selected NPIs fit within the problem types.

Response: Revised as suggested. Per NPI, we have added information on how they fit within the problem type (page 6 line 141-160):

‘’A = Nightly curfew (NC). This NPI was implemented from 23 January to 28 April 2021 in the Netherlands, as the outbreak situation was highly concerning. Much backlash from citizens arose as it highly interfered with values like freedom and autonomy. Moreover, scientific knowledge regarding the effectiveness of NCs was scarce (27). Due to these characteristics, NC was categorized as type A.

B = Closure of elementary schools and daycares (CED). This NPI was implemented 16 March and 6 April 2020 in the Netherlands, as part of the intelligent lockdown. CED was categorized as type B, as it was unknown how CED would impact the epidemic. However, many citizens expressed concerns about the safety of teachers and daycare employers and children when keeping schools and daycares open. Hereafter, the government decided to employ the CED (27).

C = Digital Covid Certificate for events (DCC). At the start of July 2021, during a very short period, this NPI was implemented. It was more or less clear that such a certificate would decrease transmission, however, there was much public debate regarding this NPI. In particular, debate about the division in society on vaccination status (28). Therefore, DCC was categorized as type C. 

D = 1.5meter social distancing (1.5M). Since March 2020 a number of basic measures were implemented. One of these basic measures was the 1.5 meter social distancing. Social distancing was an evidence-based measure to reduce transmission, and was generally well accepted by citizens (29). Due to these characteristics, 1.5M was categorized as type D.’’ 

Comment 11: Lines 133-135: This needs wordsmithing for clarification

Response: Revised as suggested, we have rephrased this and added extra information about the framework of Hoppe and Hisschemöller for clarity (page 5-6 line 133-140):

‘’Four NPIs implemented in the Netherlands were used as a starting point, and were based on the framework by Hoppe and Hisschemöller 1993 depicted in Fig 1 (26). In this framework, four problem types are described. Hoppe and Hisschemöller define a problem as a construct, that comprises both values and facts. These values and facts stand at the root of how a problem is categorized. As depicted in Fig 1, on the y-axis the level of certainty about relevant knowledge about the problem is represented. On the x-axis, the level of certainty about relevant norms and values is represented. In this study, four NPIs are identified and categorized based on the framework. By using four NPIs that fit the four problem types, information regarding a wide variety of NPI-types could be raised.’’

Results

Comment 12: In general, when discussing the different topics within each answer, I suggest keeping consistent with how they are formatted within the results section. In some instances the topics are italicized when discussed (Line 231: Inform), sometimes they are italicized and within single quotation marks (Line 241: ‘Trade-offs between interests’), and sometimes they are just within single quotation marks (Line 242: ‘Practicability’). Standardizing how these are presented will enhance readability and make it clear to the reader that a topic within a theme is being discussed.

Response: Revised as suggested, we have solely italicized each topic.

EX: Line 248-249: “during the outbreak, when it became clear that the NPI was necessary” may read as two choices when it is in fact one choice.

Response: Revised as suggested, we understand the confusion. The focus of this section is the preferences of respondents towards timing of engagement, therefore we decided to delete the second part of the sentence and solely focus on timing in the outbreak to improve comprehensibility (page 12 line 281).

EX: Line 241: ‘Trade-offs between interests’ vs. Line 242: ‘Practicability’

Response: Revised as suggested (page 12 line 270-274).

Comment 13: Lines 201-207: Please make sure the percentages as they are listed in the text align with what is demonstrated in Table 3 – Many percentages do not match up. Please fix as necessary.

EX:

- Line 202: CED, 19% did want to engage

- Line 203 DCC, 30% did want to engage

- Line 205-206: who did want to engage in decision making between ... CED (30%), DCC (19%)...

- According to Table 3: CED – 19%; DCC – 30%

Response: Revised as suggested, thank you for identifying this (page 10 line 223-226).

Comment 14: For evaluations of the Likert scales, I suggest adding a brief sentence before explaining the results to describe how you interpreted the mean values of the respondents (like did you set the midway point as 2.5 and thus any mean score above that would be agreeable vs. below that would be not agreeable?). This simple introductory sentence will help your future readers who may not be as familiar with how such data is interpreted.

Response: Revised as suggested. We have taken this up when presenting results of variables that were Likert-scale questions. First, for the results of the variables reasons to engage and reasons not to engage (page 10 line 234-235):

‘’When the mean value of this variable scored a 3 or higher, it was interpreted as a true reason for respondents (not) to engage.’’ 

And for the results of the variable about phase in decision-making process (page 12 line 268-269):

‘’When the mean value of this variable scored a 3 or higher, it was interpreted as the respondents having a positive disposition towards engagement.’’

And for the results of the variable about incorporation of contributions of citizens (page 13 line 300-302):

‘’Furthermore, for all four NPIs, respondents believed that the incorporation of contributions of citizens should be kept voluntarily, with mean values between 2.6 and 2.8 (using a score of 3 as a cut-off point between voluntary and mandatory incorporation).’’ 

Lastly, for the results of the variable about responsibility, the exact descriptions of the points on the Likert-scale are used to interpret the results (1=no responsibility at all, 2 = A little responsibility, 3 = medium responsibility, 4 = a lot of responsibility, 5=complete responsibility).

Comment 15: Lines 230-235: I suggest you should also comment on the results (as shown in Table 3) that demonstrate the least suitable mode of engagement being Empower. This shows that in general, the public just wants more information concerning the NPI decision-making process, but does not think that the decision-making process should be put in the hands of the general public.

Response: After careful consideration, we have decided to alter the presentation of our results regarding this theme. We decided to present the results in the same manner as presented in the survey. This means that we present the most chosen modes of engagement in the five ranks. In this way, we hope it is more clear for the reader to understand that Inform is ranked by the majority of respondents as most suitable mode of engagement (rank 5), and Empower ranked by the majority as least suitable mode of engagement (rank 1). 

Furthermore, now we also present the most suitable modes of engagement on rank 2,3 and 4. We also added the percentage of respondents that ranked the mode of engagement on designated rank (page 14 Table 3):

Table 3. Summary table of results for theme 1,2,4 and 5. data is presented as proportions (%), mean scores on Likert-scales (with standard deviations).

QUESTION (ANSWER) CATEGORIES NIGHTLY CURFEW CLOSURE OF ELEMENTARY SCHOOLS AND DAYCARES DIGITAL COVID CERTIFICATE FOR EVENTS 1.5M SOCIAL DISTANCING

PERCENTAGE OF ALL RESPONDENTS

1. DESIRE FOR ENGAGEMENT YES 27% 19% 30% 21%

 NO 49% 59% 46% 52%

 I DON’T KNOW / NEUTRAL 24% 22% 24% 26%

MOST CHOSEN MODE OF ENGAGEMENT IN RANK* (% OF ALL RESPONDENTS)

3. SUITABLE MODE OF ENGAGEMENT (1=LEAST, 5=MOST) RANK 1 EMPOWER (72%) EMPOWER (71%) EMPOWER (70%) EMPOWER (73%)

 RANK 2 COLLABORATE (44%) COLLABORATE (43%) COLLABORATE (45%) COLLABORATE (46%)

 RANK 3 ADVICE (47%) ADVICE (49%) ADVICE (47%) ADVICE (50%)

 RANK 4 CONSULT (41%) CONSULT (44%) CONSULT (42%) CONSULT (44%)

 RANK 5 INFORM (46%) INFORM (49%) INFORM (46%) INFORM (50%)

We have altered the text in the discussion accordingly (page 16 line 347-362):

‘’Inform was judged the most preferred mode of engagement, and was defined as receiving all information about the considerations regarding the final decision about implementing the NPI. This finding could signify a need among the public for more transparency and insights into the considerations made during COVID-19 management. Such transparency could positively influence preventive behavior during a pandemic and trust in authorities (39). The other half of respondents preferred the other modes of engagement as suitable, for example, Consult was mostly placed as the second most suitable mode of engagement, followed by Advice as the third most suitable. This reveals a desire among citizens to have more impact on decisions, rather than only being informed appropriately. Moreover, our respondents indicated that, during PE, policy-makers should carefully consider contributions of the public, but should not be obliged to adopt them. If not adopted, however, it is crucial for them toto explain why. Similarly, Litva et al (2002) identified that incorporating public perceptions in healthcare decision-making should be kept voluntarily. Additionally, they identified that when contributions are not adopted, it is vital to explain why (40). Empower was considered the least suitable mode of engagement. This corresponds with our other results that indicate citizens prefer experts and policy-makers to have more responsibility than citizens, and that contributions of citizens should not be mandatorily incorporated into decision-making.’’

Furthermore, we have added a supplementary file (Supplementary file 5), which pertains all results on suitable mode of engagement. In this supplementary file, per NPI, the percentage of respondents that ranked the five modes of engagement from least suitable to most suitable are displayed.

Comment 16: Also who was answering the “least suitable mode of engagement” question? The people who said they did want to engage or did not want to engage? This needs to be made clear.

Response: Revised as suggested, we have clarified this by specifying that the survey questions of themes 2-5 were answered by all respondents (page 11 line 256):

‘’2. Mode of engagement

The survey questions of themes 2-5 were answered by all respondents.’’

Furthermore, in the results sections about the reason to engage and reasons not to engage, we have added two sentences that these questions were only answered by respondents who did want to engage, and by respondents who did not want to engage (page 11 line 233-234 and 245-246):

‘‘Fig 2 displays the findings regarding the reasons why respondents desired to engage. The questions on reasons to engage were only answered by respondents who did want to engage (routing question).’’

‘’On the right side of Fig 2, the reasons why respondents desire not to engage are displayed. The questions on reasons not to engage were only answered by respondents who did not want to engage (routing question).’’

Comment 17: Line 258: Should “Organizations with interest in NC” be changed to “Organizations with interest in CED” instead?

Response: Yes, Revised as suggested (page 13 line 291).

Comment 18: You do not discuss the results of 5. INCORPORATION. I would discuss it (even if briefly) or remove it from Table 3.

Response: Revised as suggested. We have added this in the results section at the end of theme 5. Responsibilities (page 13 line 300-302):

‘’Furthermore, for all four NPIs, respondents believed that the incorporation of contributions of citizens should be kept voluntarily, with mean values between 2.6 and 2.8 (using a score of 3 as a cut-off point between voluntary and mandatory incorporation).’’

Comment 19: For the “Phase in Decision-making process” – you report a neutral-to-negative response. Could this be due the majority of respondents indicating that their preferred mode of engagement is “Inform” while the survey questions in the “Phase in decision-making process” are phrased in ways that the public appears to have a more active role rather than just being informed? (i.e. “engaged in assessing the severity of the outbreak” “engaged in determining the effect”)

Response: Revised as suggested. We have integrated this our discussion, in the section about the respondents that expressed no desire to engage (page 17 line 370-379):

‘’In addition, most respondents indicated that they do not want to engage (or had neutral dispositions towards engagement) in any of the phases of the decision-making process. This could be explained by the majority of respondents only wanting to be informed instead of having a more active role in e.g. assessing the severity of the outbreak situation. These results raise the question of how many citizens are required to engage in order to justify implementing PE in practice (and conversely, how many citizens should not want to engage to justify no implementation). Moreover, besides ‘’group size’’, other considerations could also be important, if not more important, regarding PE. Other considerations could include be how much impact decisions have on citizens, available resources such as time, and the need for diverse perspectives, as well as the exact payoff or impact of PE in public health.’’ 

Comment 20: Lines 277 and 279: Can you frame the likelihood statistics for preferring engagement among age groups in the inverse? (i.e. 65+ year old respondents were 3.22 times more likely to prefer engagement over no engagement in comparison to 18-24 year olds). It says the same thing, but the fold increase appears more meaningful in this manner.

Response: Thank you for this suggestion, we have inversed the two OR’s (page 14-15 line 307-312):

‘’ Furthermore, for all four NPIs, age also appears to be a predictor for desire to engage; the higher the age group of the respondents, the more likely they would want to engage in decision-making. For DCC for example, 65+ year old respondents were 3.23 (p<0.01) times more likely to prefer engagement over no engagement compared to 18-24 years old. Similarly, when comparing 65+ years old to 25-34 years old, the older group was 2.38 (p<0.01) times more likely to prefer engagement over no engagement.’’ 

Comment 21: Have you done any comparative statistics concerning the demographic variables and the survey themes/topics? If so, were there any findings of interest? I wonder if some of these demographic variables also have an impact on the survey responses.

Response: Thank you for raising this point. Looking into differences in the responses of citizens on engagement questions based on demographic characteristics was not a primary aim of our study, as we limited our scope to the preferences regarding engagement among the general public as a whole. To provide some insights into variations in the responses among the general public, we decided to add the multinominal regression analysis on the variable of desire for engagement (which could be seen as a basic preference, relating to all other questions in the survey). We decided not to repeat this analysis for all outcome variables in our study, as this would mean a large number of additional analyses leading to data-driven outcomes rather than to answers to our research question. However, we do see the value of emphasizing how much variety there is within ‘’the public’’. Furthermore, we think it would be worthwhile to gain more in-depth knowledge regarding the preferences of various population groups towards PE in pandemic management. Our study was a first step in gaining preferences, and a next step would be to differ between population groups. This differentiation could be based on demographic characteristics, but also in the relationship of citizens towards the problem. We have added this to our limitations (page 18 line 405-410):

‘’ As a final point, the public is comprised of people with a diverse range in characteristics. In this study, we have taken a first step in gaining insight into the preferences of the public. However, it would be valuable to gain more in-depth insight into differences in preferences in various groups within the public. This variety could be explored through demographic characteristics, as well as positions on or viewpoints about the COVID-19 pandemic; for example, this could be done by collecting data on the views of vulnerable populations or people with a lack of trust in the government.’’

Discussion

Comment 22: Line 314: Replace “extra” with “additional”.

Response: Revised as ‘’PE’’ instead of ‘’This extra perspective’’ (page 15 line 336).

Comment 23: Line 312-315: Expand on what you mean by valuable (EX in the sense of government trust, understanding of NPIs, etc.). Also would it be new insights? Your survey results say that most respondents just wanted additional information, not necessarily to provide input. If you re-phrase this in the sense of the results suggesting that provision of more information concerning the NPI design and execution could be valuable for the public, I think that would be more in agreement with your results.

Response: Revised as suggested. We have rephrased the sentence and give an explanation about in which sense PE could be beneficial. We have focused on informing the public as this was the most suitable mode of engagement. However we also added consulting the public as we also found that around half of our respondents deemed more active modes of engagement (consult to empower) as most suitable (page 15 line 332-336):

‘’The results suggest that insights from the public could be beneficial, especially for NPIs for which there is no public consensus . For such NPIs, PE could have the benefit of informing the public and improving understanding of the implications of NPIs. Furthermore, a more active mode of engagement like consulting the public, could have benefit in improving the quality of NPIs.’’ 

Comment 24: Line 315: “made under high-pressure time constraints”

Response: Revised as ‘’made under time pressure’’ (page 16 line 337).

Comment 25: Line 317-319: Your data provides evidence for this, as the mean value for CED “no direct impact of NPI” within the reasons to not engage section of Figure 2 is 3.9. I would refer to this here.

Response: Revised as suggested. We indeed meant to conclude this based on our findings. We have revised the sentence to make it more clear (page 16 line 338-340):

‘’For CED, this can be explained by the lack of direct impact on daily life, as not all citizens have (pre) school-aged children who attend elementary school or daycare.’’

Comment 26: Line 331-332: Where is this shown? Is this the interpretation of the results from 5. INCORPORATION? If so, you for sure want to make sure that this is discussed in the results section, otherwise these results are coming out of nowhere, without any previous discussion or context.

Response: Revised as suggested. We have added the findings on the item Incorporation in the results (page 13 line 300-302):

‘’Furthermore, for all four NPIs, respondents believed that the incorporation of contributions of citizens should be kept voluntarily, with mean values between 2.6 and 2.8 (using a score of 3 as a cut-off point between voluntary and mandatory incorporation).’’

Comment 27: Line 333-335: At the same time, you have results that say over 70% of respondents say empower is the least suitable mode of engagement... I think these results are more telling of public opinion, and is more in line with the general support for experts and policy-makers in the decision-making process. It might be worthwhile to comment on both findings (i.e. empower being the most important mode after Inform, but empower also being voted the least appropriate mode).

Response: Revised as suggested. We have added the findings on Empower being the least appropriate mode of engagement according to our participants in the discussion (page 16 line 359-362):

‘‘Empower was considered the least suitable mode of engagement. This corresponds with our other results that indicate citizens prefer experts and policy-makers to have more responsibility than citizens, and that contributions of citizens should not be mandatorily incorporated into decision-making.’’ 

Comment 28: Line 346: “A quarter of... desire to engage” I would remove this sentence – it does not really add anything to the discussion where it currently is.

Response: Revised as suggested, we have deleted this sentence.

Comment 29: Line 347-351: I think another thing to consider is how PE is expected to impact/benefit NPIs. For example, while you have increased understanding of NPIs and trust of government, but among those that did want engagement, you had relatively neutral mean scores from respondents regarding increased acceptance/adherence of NPI. So if you put all this effort into public engagement, but without the public health payoff of increased NPI adherence, would PE actual be beneficial?

Response: Revised as suggested, we have added this consideration in this part of the discussion (page 17 line 370-379):

‘’In addition, most respondents indicated that they do not want to engage (or had neutral dispositions towards engagement) in any of the phases of the decision-making process. This could be explained by the majority of respondents only wanting to be informed instead of having a more active role in e.g. assessing the severity of the outbreak situation. These results raise the question of how many citizens are required to engage in order to justify implementing PE in practice (and conversely, how many citizens should not want to engage to justify no implementation). Moreover, besides ‘’group size’’, other considerations could also be important, if not more important, regarding PE. Other considerations could include be how much impact decisions have on citizens, available resources such as time, and the need for diverse perspectives, as well as the exact payoff or impact of PE in public health.’’

To add a little more explanation on the benefit of PE, we do believe that the added benefit of PE will payoff in more than just a better adherence to NPIs. In theory, engagement processes can add to the fairness of decisions, and could identify concerns or solutions policy-makers might overlook (as described in the introduction). The latter payoff of identifying new solutions could be scaled under the quality of NPIs. The fairness of decisions is a more overarching phenomenon that is not directly linked to individual preferences (therefore also not taken up as a reason to engage in the survey), however it could still be a payoff for more legit public health. 

Comment 30: If there are any statistics concerning NPI adherence within the Netherlands, the addition of this information would give important context for the paper and its results. This can be added to the introduction section.

Response: Revised. Indeed, there are statistics concerning NPI adherence and public support for NPIs. However, most of these statistics represent the opinions of only certain groups in society, and results of these surveys were not always in line with the tendency in the whole of society. Nevertheless, we agree that this information could provide important context, and therefore we have added (available) information regarding public support for the NPIs in supplementary file 1 regarding the information about the four NPIs. 

Limitations

Comment 31: Line 370: What is it meant by “put theory into practice when it comes to ... citizenship”?

Response: Revised as suggested. With this comment, we mean that we have tried to measure the willingness of respondents to engage in NPI decision-making by asking them their desires for engagement retrospectively. Meanwhile, it is still uncertain what the exact willingness will be to actually participate in real-time engagement efforts.(page 18 line 403-405):

‘’Another factor is the distinction between intention to engage, which we identified in this study, and actual real-time engagement when the opportunity arises. It is uncertain to what extent people are actually willing to participate in real-time engagement efforts (46).’’

---

## [Decision Letter · Decision Letter 1]

14 Sep 2023

Preferences for public engagement in decision-making regarding four COVID-19 non-pharmaceutical interventions in the Netherlands: a survey study

PONE-D-22-27762R1

Dear Dr. Kemper,

We’re pleased to inform you that your manuscript has been judged scientifically suitable for publication and will be formally accepted for publication once it meets all outstanding technical requirements.

Kind regards,

Ali B. Mahmoud, Ph.D.

Academic Editor

PLOS ONE

Additional Editor Comments (optional):

Reviewers' comments:

Reviewer's Responses to Questions

**Comments to the Author**

1. If the authors have adequately addressed your comments raised in a previous round of review and you feel that this manuscript is now acceptable for publication, you may indicate that here to bypass the “Comments to the Author” section, enter your conflict of interest statement in the “Confidential to Editor” section, and submit your "Accept" recommendation.

Reviewer #2: All comments have been addressed

2. Is the manuscript technically sound, and do the data support the conclusions?

Reviewer #2: Yes

3. Has the statistical analysis been performed appropriately and rigorously? 

Reviewer #2: Yes

4. Have the authors made all data underlying the findings in their manuscript fully available?

Reviewer #2: Yes

5. Is the manuscript presented in an intelligible fashion and written in standard English?

Reviewer #2: Yes

6. Review Comments to the Author

Reviewer #2: Upon proofreading, please make sure to go through the manuscript again. There are still instances of extra spaces and extra periods.

7. PLOS authors have the option to publish the peer review history of their article (what does this mean?). If published, this will include your full peer review and any attached files.

Reviewer #2: No

---

## [Editor Report · Acceptance letter]

28 Sep 2023

PONE-D-22-27762R1 

Preferences for public engagement in decision-making regarding four COVID-19 non-pharmaceutical interventions in the Netherlands: a survey study 

Dear Dr. Kemper:

I'm pleased to inform you that your manuscript has been deemed suitable for publication in PLOS ONE. Congratulations! Your manuscript is now with our production department. 

Kind regards, 

on behalf of

Dr. Ali B. Mahmoud 

Academic Editor

PLOS ONE